# Global data-driven prediction of fire activity

Francesca Di Giuseppe [1,2] ✉, Joe McNorton [1,2] ✉, Anna Lombardi[1] & Fredrik Wetterhall [1]

Recent advancements in machine learning (ML) have expanded the potential use across scientific applications, including weather and hazard forecasting. The ability of these methods to extract information from diverse and novel data types enables the transition from forecasting fire weather, to predicting actual fire activity. In this study we demonstrate that this shift is feasible also within an operational context. Traditional methods of fire forecasts tend to over predict high fire danger, particularly in fuel limited biomes, often resulting in false alarms. By using data on fuel characteristics, ignitions and observed fire activity, data-driven predictions reduce the false-alarm rate of high-danger forecasts, enhancing their accuracy. This is made possible by high quality global datasets of fuel evolution and fire detection. We find that the quality of input data is more important when improving forecasts than the complexity of the ML architecture. While the focus on ML advancements is often justified, our findings highlight the importance of investing in high-quality data and, where necessary create it through physical models. Neglecting this aspect would undermine the potential gains from ML-based approaches, emphasizing that data quality is essential to achieve meaningful progress in fire activity forecasting.

Advancements in machine learning (ML) have opened new possibilities in weather prediction in recent years[1-5]. The use of ML technologies in place of physically based methods to predict weather has improved the accessibility and precision of forecasts[6] and blurred the lines between physical and human-generated information. Models that adapt in real time to changing patterns of both physical variables[7] and human behavior[8,9] are now capable of exploiting the information content locked in social interactions[10,11] and has the potential to improve the prediction of human-influenced natural hazards[12]. This is especially important because humans have and still are modifying the environment on a large scale[13].

The prediction of landscape fires can benefit from this data-driven revolution. Fire is a complex process influenced by various inter-connected factors, including fuel composition, weather, and ignition[14]. Fires are predominantly human-induced in large parts of the world, making them inherently stochastic and challenging to predict. Even natural ignitions, such as those caused by lightning, present significant forecasting difficulties[15,16]. As a result, most forecast models are specific to local regions and cannot be easily applied globally[17]. Predictive models have until now relied on the concept of danger ratings. Danger ratings are empirical metrics used in current global early warning systems to mark anomalous fire weather and potential fire behavior, provided that an ignition occurs. These early warning systems use observed weather data or forecasts from numerical weather prediction models[18,19] but do not take into account variations in fuel status and abundance. Fuel information is only available on a very local scale. By not including two crucial components, fuel and ignitions, current fire forecasts only inform on anomalous weather conditions rather than provide a reliable prediction of fire occurrence. Including fuel and ignition could improve the reliability of fire risk prediction.

Excluding fuel in fire forecasts can lead to an underestimation of fire severity, for example the Alexandroupolis fire in Greece, and the extended burning in western Amazonia in 2023[20]. The unprecedented and uncontrollable urban fire that raged across Los Angeles in 2025 was in part driven by fire prone conditions, but also by an unusual accumulation of fuel from the preceding wet springs. Typically, global fire forecasts do not take into account fuel because of insufficient information on fuel availability and status. Furthermore, even with

[1]ECMWF, European Centre for Medium-range Weather Forecast, Shinfield park, Reading RG29AX, UK. [2]These authors contributed equally: Francesca Di Giuseppe, Joe McNorton. ✉e-mail: Francesca.DiGiuseppe@ecmwf.int; Joe.McNorton@ecmwf.int

good observations of fuel availability and status it would be complex to establish a physical relationship between fuel variables and fire purely through a process-based analysis.

Significant advances have been made in the ability to observe wildfires from space with the increased availability and capability of remote sensing[21–24]. Despite the challenges in detecting small fires[25] and decontaminating data from spurious signals[26], satellite data now provide a valuable global status of fire activity at increasingly detailed resolution. Satellite data have transformed the study of fire occurrence, patterns, trends, and controls, which would not have been possible using incomplete national or regional inventories[24,27–29]. In addition, global fire activity observations provide the opportunity to employ data-driven modeling to proactively address the information gap in fire ignitions.

Machine learning is already widely applied in multiple areas of fire science, including fire management practices[30], early[31] and long-term detection[32], and crucially to aid firefighting operations on a very local scale[33–35]. Global proof-of-concept AI predictive systems have mostly focused on reproducing burned areas. However, understanding the feasibility of a global, kilometer-scale fire activity prediction system is still in its early stages. An analysis of the skill of such a system, its limitations, and the key factors driving its performance is still lacking. In recent years, the European Centre for Medium-Range Weather Forecasts (ECMWF) established a system to predict the probability of wildfires occurring anywhere on the planet at least a week in advance[36]. This paper reflects on ECMWF's operational data-driven fire predictions, which have been running since 2023, and analyzes in details the extensive fire events that burned large parts of Canada in 2023 and the devastating 2025 fire in Los Angeles, California. We evaluate how skillful this completely data-centric system has been, and determine where this predictability mainly comes from-whether it is the training data or the data-driven approach itself. The key question is: do current forecasts lack skill because of the quality of input data, or because traditional physical models lack the complexity to capture wildfire processes? Understanding this fundamental issue is critical not only to

confirm trust in ML methods but also to understand the importance of good quality training data.

## Results

### Data as a source of fire prediction

The occurrence and severity of landscape fires are typically explained by the fire triangle, which considers three key factors: ignition source (either man-made or natural, such as lightning), fuel (abundance, condition, and continuity), and weather (wind, temperature, and moisture conditions)[37].

To evaluate the importance of data compared to the complexity of machine learning frameworks, three models with increasing complexity (random forest, XGBoost and neural networks)[36] were used in a set of ablation experiments, progressively incorporating additional data sources during model training (see Table 1). Designing a clear set of experiments to pinpoint the importance of each data source is challenging, particularly given the acknowledged complexity of interactions between fuel, weather, and ignition. For example, vegetation abundance and moisture levels are influenced by weather conditions, while local vegetation can also modulate weather patterns. Consequently, the contributions of fuel, weather and ignition to fire activity are not entirely independent of each other. To account for the possible combinations of these controls, we trained several data-driven models using various combinations of factors from the fire triangle (Fig. 1). This included pairing weather with fuel, fuel with ignition, and ignition with weather. Additionally, models were trained using each control in isolation and, finally, incorporating all three controls together.

Validating the prediction skill on intrinsically stochastic processes is challenging, with no suitable single metric. Therefore, we used a selection of skill scores to assess prediction skill when using different controls and how they contribute to the global prediction of fire activities (see details in the "Methods" subsection "Validation metrics"). The consistently low probability predictions (<5%) results

**Table 1 | Datasets used for training: Nineteen predictors are grouped into three categories: weather, fuel, and ignition**

| Variable | Input category | Frequency | Source | Reference |
|---|---|---|---|---|
| 2 m Temperature | Weather | Daily | ERA5-Land | 69 |
| 2 m Dewpoint Temperature | Weather | Daily | ERA5-Land | 69 |
| 10 m Wind Speed | Weather | Daily | ERA5-Land | 69 |
| Precipitation | Weather | Daily | ERA5-Land | 69 |
| Live Leaf Fuel Load | Fuel | Daily | Fuel model | 38 |
| Live Wood Fuel Load | Fuel | Daily | Fuel model | 38 |
| Dead Foliage FuelLoad | Fuel | Daily | Fuel model | 38 |
| Dead Wood Fuel Load | Fuel | Daily | Fuel model | 38 |
| Live Fuel Moisture Content | Fuel | Daily | Fuel model | 38 |
| Dead Foliage Moisture Content | Fuel | Daily | Fuel model | 38 |
| Dead Wood Moisture Content | Fuel | Daily | Fuel model | 38 |
| Vegetation OpticalDepth | Fuel | Monthly | Satellite (SMOS) | 82 |
| Low Vegetation LAI | Fuel | Monthly | Satellite (multi- sensor) | 83 |
| High Vegetation LAI | Fuel | Monthly | Satellite (multi- sensor) | 83 |
| Type of Vegetation | Ignition | Fixed | ECLand | 83 |
| Urban Fraction | Ignition | Fixed | ECLand | 84 |
| Orography | Ignition | Fixed | ECLand | 83 |
| Lightning | Ignition | Daily | ERA5 | 85 |
| Population Density | Ignition | Fixed | Gridded population of the world (GPW) v4-SEDAC (2020, 2.5 arcmin to 9 km) | Center for International Earth ScienceInformation Network- CIESIN |
| Road Density | Ignition | Fixed | Global Roads Inventory Dataset- 2018 | 86 |

The ablation experiments are constructed using progressively larger sets of data, as grouped in the table presented.

## Quality of the fire predictive model

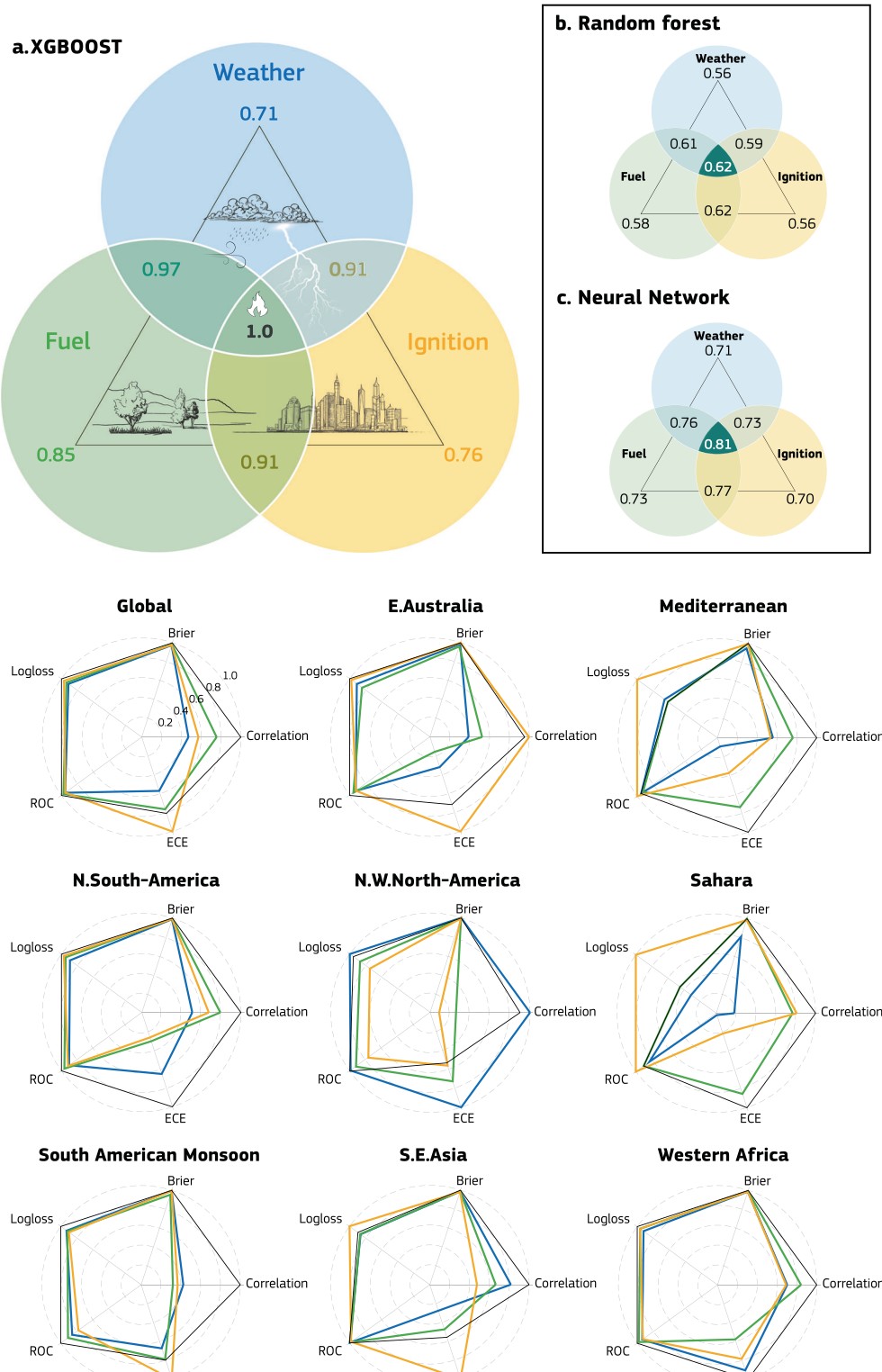

in a strong penalization in probabilistic scores like Brier and Logloss even when fire events occur, as these metrics are designed to penalize the distance from a deterministic-like prediction of 100% probability in case of a hit. To provide a generalized assessment, we averaged across all skill scores, even recognizing that they are different in nature and not uncorrelated. The best-performing ML architecture is the intermediate-complexity model, XGBoost, which

significantly outperforms the simpler random forest while offering performance equivalent to the more complex neural network. This indicates that a deep layered infrastructure does not provide additional accuracy for classification problems such as fire ignitions. Combining all data provides the best prediction of fire activity for all infrastructures, both globally and regionally. From an ideal starting point where all sources of information are included, a 30%

**Fig. 1 | Skills of the data-driven fire prediction.** In the upper panel, an info-graphic summarizes the average skill score across a wide array of metrics used to evaluate the performance of data-driven models derived from different combinations of training data (see the "Methods" subsection "Validation metrics"). The XGBoost architecture (panel **a**) is the best-performing model, despite being of mid-level complexity. While XGBoost outperforms a random forest (panel **b**), increasing complexity (e.g., using a neural network (panel **c**)) does not lead to significant gains. Therefore please note that all scores are presented relative to XGboost performance represented as a 1 at the center of the main triangle. Combining all data sources yields the best fire activity predictions, both globally and regionally. From this ideal scenario, prediction skill degrades by roughly 30% when using only weather or ignition data, and 150% with only fuel data (for the XGboost). Similar decreases are obtained also with the other architectures. Fuel data is especially important, as it captures the effects of weather on vegetation. Using any two data sources improves prediction quality, reducing the degradation to between 17% and 13% relative to only using one source. In the lower panels, spider plots for the XGboost model provide detailed breakdowns of individual metrics for selected regions among the IPCC AR6 WGI reference regions (see Supplementary material). There are regional differences in which weather (e.g., northwestern North America) or ignitions (e.g., eastern Australia) play a major role in the quality of predictions. However, overall, fuel remains the most relevant factor globally.

degradation of skill is expected when only weather or ignition is considered, and 15% when only fuel is considered. This is not surprising as fuel includes weather-driven variables like fuel moisture, making it the most relevant factor. Any combination of two controls increases the quality of the prediction, resulting in a degradation of only 3–7% of the maximum achievable skill. The improvement achieved by incorporating additional data into the training process outweighs the gains obtained from transitioning from a medium-complexity to a high-complexity architecture.

There are regional differences, especially where fire activity is inhibited (Sahara) or mostly driven by lightning ignitions (eastern Australia)[15]. In the latter ignitions play a relevant role, and even our simple representation based on lightning forecasts and static maps of population and road density provides the best possible input for a fire forecasting system. Regions for which traditional fire weather indices were designed, such as northwestern North America, still select weather as the most relevant control to explain fire activity, confirming the weather-limited regime of forested areas. However, the fuel data derived through the physical model of McNorton and Di Giuseppe[38] is the most relevant control in isolation, not only in fuel-limited ecosystem such as the Mediterranean region and western Africa, but also in tropical regions in South America and globally. Fuel is here represented as a combination of fuel abundance and fuel moisture content, and changes are mainly driven by weather condition and human influence on land use and land cover. Fire activity in regions controlled by fuel account for the bulk of fire activity globally and are the largest source of carbon emissions (e.g. ref. [39]).

## Predicting fire activity rather then fire danger

One of the most significant limitations of traditional fire weather indices is their tendency to consistently predict high fire danger in regions with low fuel availability[40]. For example, FWI often indicates extreme fire danger in desert areas where extreme temperatures and low moisture persist for most of the year, despite the absence of fires due to insufficient vegetation. Such areas are typically masked out in applications using fixed land-cover datasets. In fact, when comparing the climatology of fire danger for 2023 with the actual fire activity recorded for that year, the FWI highlights extreme values in vast desert regions such as the Sahel, the Tibetan plateau, and the Gobi desert, where no fire activity was recorded (Fig. 2). A data-driven approach trained directly on observed fire activity forecasts fire activity rather than fire danger, which to some extent addresses this issue. The model avoids the false predictions characteristic of traditional indices by learning the locations of barren areas from the climatological absence of fire activity (Fig. 2). Importantly, this learning process is significantly improved by expanding the training dataset. When weather variables are the sole input, the inhibitory factors related to the lack of recorded fire activity stem only from the target dataset. However, incorporating additional information on fuel availability and ignition sources during training enables the forecasting system to better reproduce the global climatology of fire activity. This underscores the substantial predictive improvements that can be achieved by integrating relevant data during the training phase.

## Predicting extremes

Another interesting aspect is the prediction of extremes and model confidence when making high probability predictions (Fig. 3). As fires are sporadic events subject to favorable conditions but ignited through stochastic triggers, they are systematically over-predicted in deterministic process-based forecasting systems (also visible in Fig. 2). The inherently probabilistic nature of the data-driven approach is therefore advantageous. In this logical framework, accuracy means that the predicted probability should match the frequency of the events, a metric typically referred to as the reliability of the forecast. The frequency of observing a single fire in any given year within a 9 km grid box is expected to be low (1 time in 365 days would lead to an average ~0.3% probability for each day), so we also expect a low probability to be the correct forecast. At low probabilities, all data-driven models perform equally well globally with a prediction probability that matches the observed frequency (Fig. 3). However, the weather-only data-driven model tends to over-predict the forecasts for high probabilities. Including information on fuel availability and sources of ignition not only substantially improves the predictions in many regions, but also increases the model confidence in predicting higher probabilities. This becomes a key aspect when these systems are considered for operational early warning systems. For example, a false alarm could result in investing resources when not needed and a forecast miss could result in more severe consequences, such as not issuing warnings and a lack of preparation for wildfire events.

## Real-time application for a data-driven fire prediction system

The use of data-driven prediction for fire activity is showcased in two events: one in California in 2025 and another in Canada in 2023.

Starting on January 7, 2025, a series of catastrophic wildfires devastated the Los Angeles metropolitan area and surrounding regions. These fires were fueled by extremely low humidity, dry conditions, and hurricane-force Santa Ana winds, which reached speeds of 160 km/h in some areas. The wildfires claimed several lives, destroyed or damaged thousands of residential and commercial structures, and forced nearly 200,000 people to evacuate. The most severe impacts were caused by the two largest fires: the Palisades fire and Eaton fire, which remained uncontrolled for days. Traditional fire weather indices had indicated that persistent anomalous and dry conditions for the winter season and the incoming katabatic winds would create ideal anomalous conditions and warnings had been issued over the regions. However, FWI indicated widespread danger conditions (Fig. 4). The data-driven model including all the variables shows better localization of areas where fires ignited (Fig. 4). This better skill stems from the inclusion of information on fuel abundance and status as well as proximity to human activities. In fact, the severity of the fires was linked to a phenomenon known as "hydroclimate whiplash"[41] which is believed to be exacerbated by climate change. This phenomenon occurs when anomalous wet conditions are followed by very dry conditions. The wet spell promotes vegetation growth and a rapid increase in fuel availability. During the subsequent dry period, moisture is quickly depleted, leaving the fuel highly flammable and often resulting in severe burning. California experienced two very wet spring

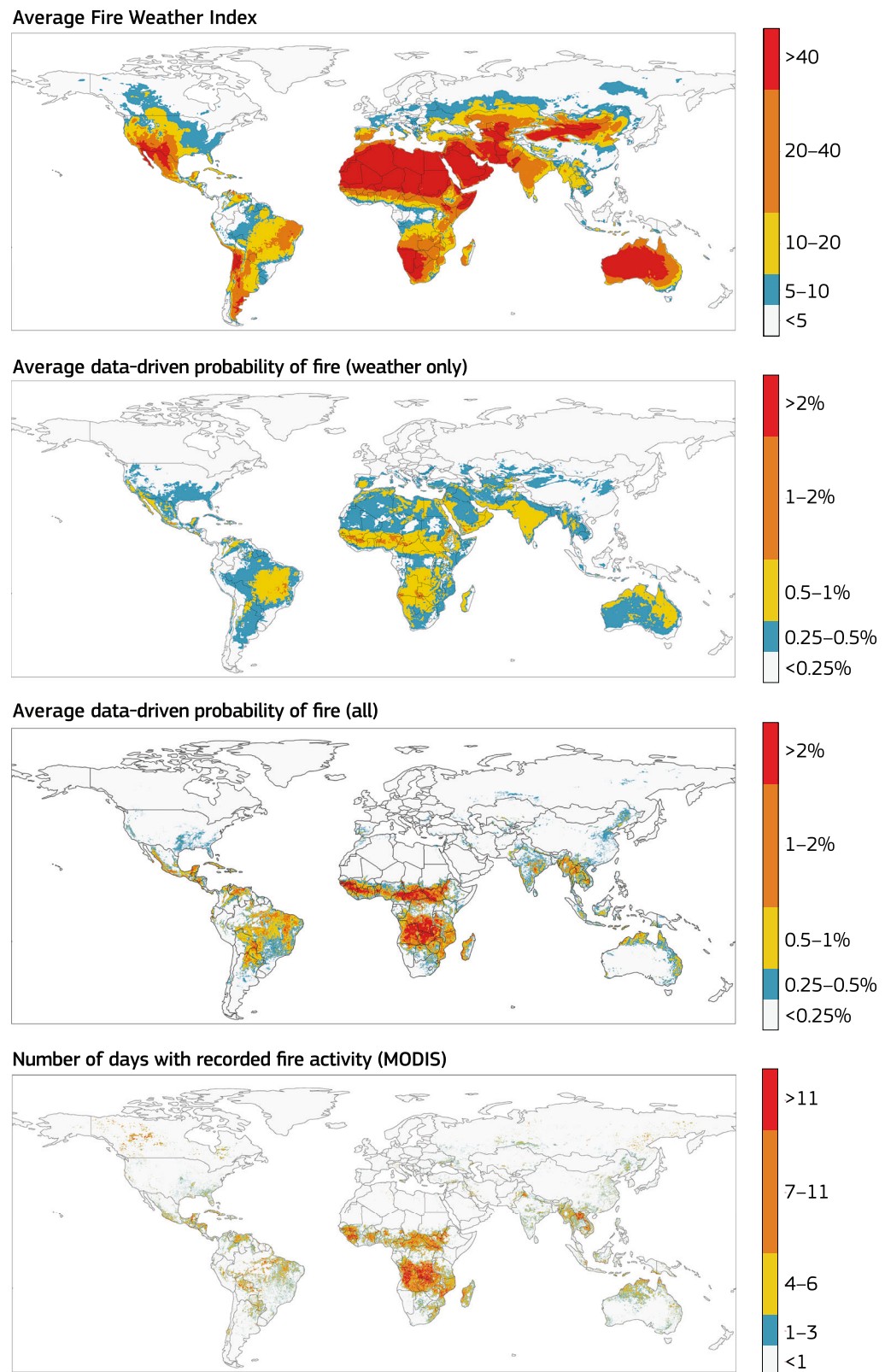

**Fig. 2 | Predicted yearly climatology of data-driven and fire weather indices.** The top figure shows the yearly average of the fire weather index[87] with values color-coded according to the current recommendations of the Global Wildfire Information System[19]. Middle panels display the annual climatology for the data-driven probability of fire driven by weather conditions only and driven by all controls (weather, fuel and ignitions). The last panel shows the recorded fire activity in terms of active fires for 2023. It is evident how the inclusion of multiple controls allows for the complete elimination of barren areas among regions with a high probability of fires. It also enables the clear identification of areas where fires are recurrent processes (e.g., sub-Saharan and Central Africa) and where the number of fire events is more than one per year. As more data are included to train the model, the prediction becomes more confident and closer to the observed fire activity on Earth.

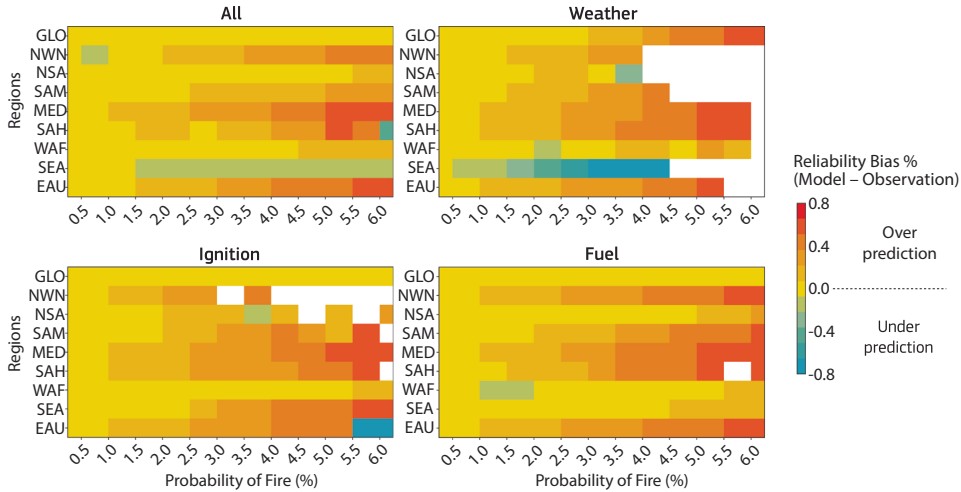

**Fig. 3 | Reliability of the data-driven fire prediction.** Reliability bias (Model-Observation) is plotted for different forecast probability predictions, ranging from low to high probabilities of fire. The data-driven predictions are probabilistic, meaning accuracy is determined by how well the predicted probabilities match observed event frequencies (see the "Methods" subsection "Validation metrics"–"Reliability"). For reference, in a 9 km grid box, the probability of a fire occurring on any given day is -1/365, or a 0.3% probability per day, assuming all days are equally likely. Thus, very low probabilities are often expected to be the correct forecast. The weather-only data-driven model tends to over-predict high probabilities due to a lack of fuel availability data, but incorporating fuel and ignition information significantly improves predictions and increases model confidence, which is crucial for operational early warning systems to avoid costly false alarms or, worse, missed warnings leading to inadequate wildfire preparedness.

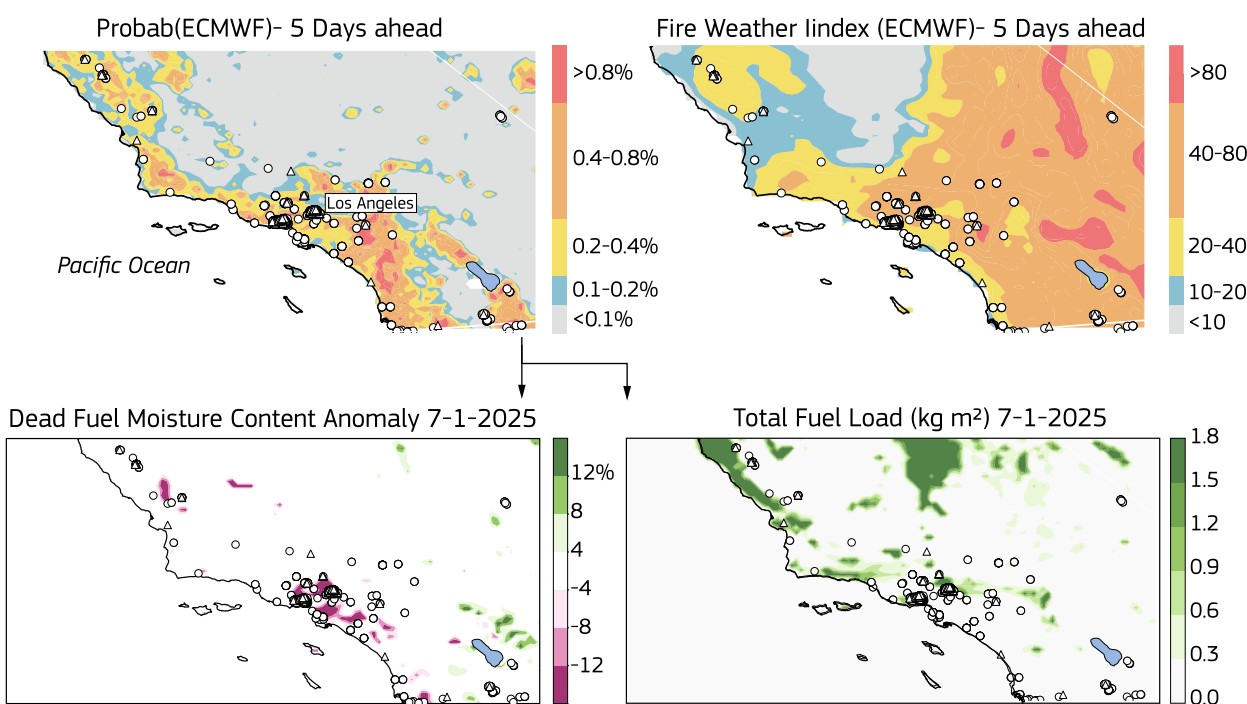

**Fig. 4 | California data-driven fire prediction for 2025.** Five days forecast leading to the 7 January 2025 when major fire outbreaks affected the Los Angeles metropolitan area. The data-driven model including all information of fuel status shows better localization of areas where fires ignited. California experienced two unusually wet spring seasons leading up to the 2025 fires, resulting in an increased accumulation of fuel in the region. These anomalies led to a unique pattern of heightened flammability at the wildland–urban interface. The data-driven model successfully identified this pattern by accounting for the abnormal fuel conditions.

seasons preceding the 2025 fires, which contributed to an enhanced availability of fuel in the region. These anomalies created a distinct pattern of increased flammability at the wildland–urban interface. This pattern was effectively captured by the ML model, which accounted for the anomalous fuel conditions but remained undetected by traditional FWI metrics.

The 2023 fire season in Canada began earlier than usual, mirroring a recurring trend of spring fires in the northern prairie provinces attributed to early spring drying and the emergence of overwintering smoldering fires from warmer winters[20,42,43]. Notably, British Columbia witnessed its first wildfire evacuation of the season in mid-April, while Nova Scotia's capital Halifax experienced the largest wildfire on record, prompting the evacuation of over 16,000 individuals in late May[44] (Fig. 5 upper panels). In June, Quebec encountered two lightning outbreaks, igniting hundreds of new fires, and resulting in a record-high burned area of 4300 km² [20] (Fig. 5 lower panels).

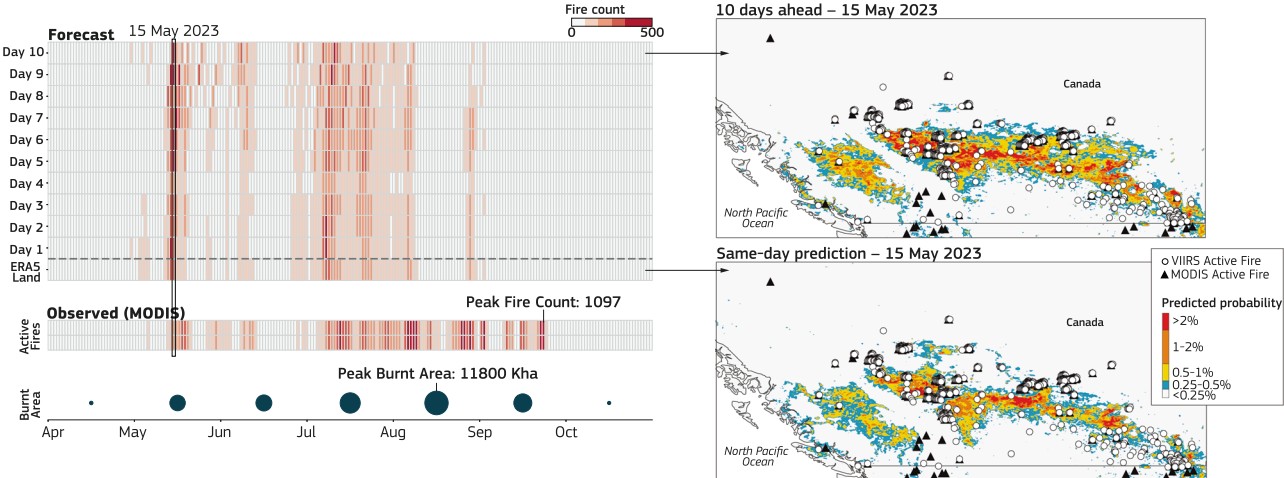

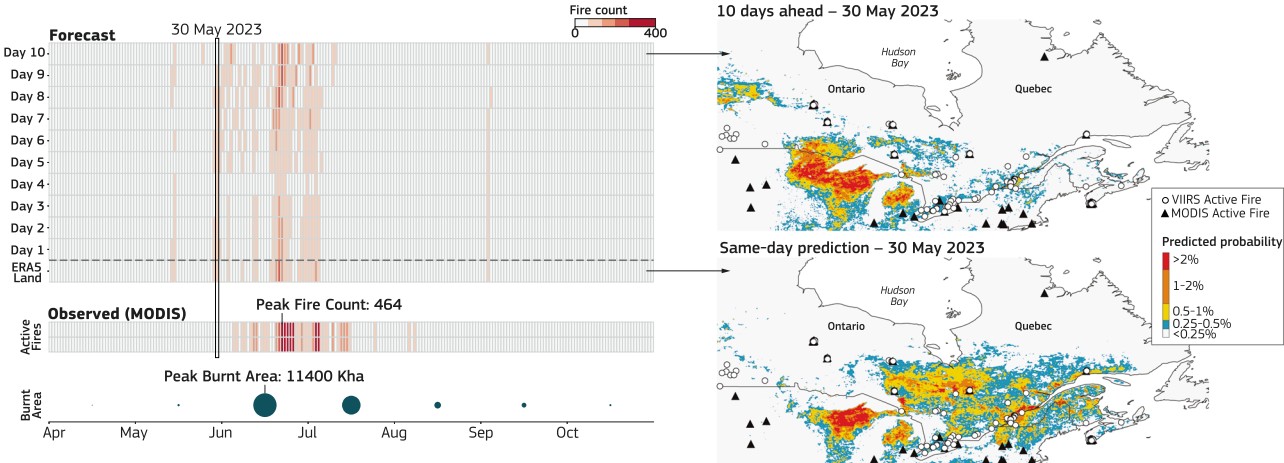

**Fig. 5 | Canada data-driven fire prediction for 2023.** Left: A chiclets plot displays fire predictions over time in terms of the number of forecasted fires. For observations, this is based on the daily count of MODIS active fires in the region. For forecasts, it is calculated by summing the probabilities across the area. The x-axis corresponds to specific dates throughout the year, while the y-axis represents either the observations or the forecast lead time (often referred to as the forecast horizon). The vertical color coherence allows for quick identification of the time windows of predictability associated with observed fire activity. The total burned area per month (circles) is shown as a reference to indicate the severity of the fires, though it is not used in the prediction. Right panels: Predictions from the data-driven fire model for May 13, 2023, in Western Canada, and May 30, 2023, in Eastern Canada, with superimposed MODIS and VIIRS active fire detections on those days. The areas shown on the maps correspond to those used to generate the chiclets plots.

Given the Canadian wildfire season in 2023 was significantly more widespread than any seen in our training data, we hypothesize if a data-driven approach can perform under conditions it was not trained for, and if it can be useful for real-time applications? Our analysis shows that both versions of the forecast model, a standard 9 km version presented here and an experimental 1 km version, deliver accurate information of extreme fire activity for the Canadian fires when trained using all available drivers, offering valuable insights up to 10 days in advance (Fig. 5).

Both the spatial distribution and the total intensity of fire activity is well forecasted 10 days ahead when we compare to recorded MODIS active fires. The model correctly predicts fires even when MODIS is unable to detect them because of cloud or smoke. MODIS data used in our analyses are known to be conservative because of limitations in detecting small fires based on surface spectral changes at 500 m resolution[24]. We showcase a typical event on 30 May 2024 where MODIS detection is underestimated, and measurements from other sensors (VIIRS) depict more intense fire activity (Fig. 5 lower panel).

Since our model is trained globally, and the missing observations from MODIS are not systematic, the data-driven approach can correct for these satellite omission deficiencies.

This capability is relevant not only for early warning systems but also for informing on possible missing observations. Emission forecasting systems relying on observations of fire radiative power are known to underestimate the contribution of fire emission to air quality forecasts[45]. Despite most of the Quebec fires were in remote regions, the smoke they generated blanketed several major cities in eastern North America, including New York, the latter experiencing its worst air quality in half a century. The observed daily mean PM2.5 concentration rose to 148.3 µm$^{-3}$ on 7 June 2023, over four times the recommended daily limit[46]. In total, over 50 million people were exposed to high levels of PM2.5 for several days[47]. The full extent of the fires over this period were missing from MODIS observations resulting in observation-based emission estimates (e.g. The Copernicus Atmosphere Modeling System) severely underestimating the total amount of PM2.5.

## Discussion

Since the 1970s, landscape fire predictions have relied on empirical models of landscape flammability tailored to specific ecosystems[48–50]. They have become pivotal tools for fire management agencies in preemptively identifying critical areas for suppression[51,52]. The ease of implementation and the availability of weather data have contributed significantly to their widespread use. Despite their utility, studies have highlighted the limited effectiveness of the fire weather index and similar metrics in fuel-limited ecosystems where fires are driven by the short-term superficial drying of intermittently available biomass[40]. The availability of remote observations for fuel, either independently[53] or supported by modeling frameworks[38,54,55], has permitted the development of new indices that partially incorporate fuel considerations into their formulation[40,56]. However, it is the emergence of data-driven technology that holds the promise of significantly enhancing our predictive capabilities[36] as it allows us to exploit information from diverse sources in a computationally efficient way. Using a set of ML infrastructures we have shown that a data-driven approach of fire prediction can:

- learn conditions where fire is inhibited, even when only utilizing the same atmospheric variables as the FWI.
- Rapidly integrate information on fuel without the complexity of formulating a process-based framework connecting fuel to fire activity.
- Consider ignitions that may lack a direct physical basis, e.g, when induced by humans.
- Adapt and refine its predictions over time in a simple way, allowing it to learn about changes in flammability due to climate change[57] or human practices[58,59]. Updates in process-based derivation would require more targeted data acquisitions (e.g., through prescribed burnings), which are more difficult to obtain and site specific.

A data-driven model can forecast the probability of fire activity itself, unlike process-based methods. This enhances the usability of data-driven outputs as they directly relate to observable variables of fire activity. The quality of the forecast can be verified in a probabilistic framework against a measured quantity. This opens new avenues in our capability to forecast fire activity, as it becomes feasible to generate an initial estimate of active fire observations, such as those from MODIS. This could allow for the correction of missing observations, which would have a substantial impact if applied to fire emission estimations which are highly affected by availability of satellite observations[60–62].

A data-driven approach is likely to outperform process-based methods in most circumstances, but the effectiveness of data-driven predictions relies on the quality and relevance of the input data. Data-driven models can still encounter challenges, such as over-predicting fire activity in sparsely vegetated areas. This is especially true when only using the same input variables as traditional fire weather indices. Weather input alone does not provide all the information needed to constrain the problem. It is essential to consider all factors that contribute to fire activity (the three apexes of the fire triangle) during model training. Only when we incorporate weather conditions, fuel characteristics, and elements related to ignitions, we substantially improve the accuracy and reliability of fire activity forecasts. By fine-tuning the model with comprehensive input data it becomes better equipped to generate more precise predictions across various landscapes and conditions, resulting in up to 30% improvement.

Fuel status is the single most important predictor. Thus, the lack of direct real-time global observations of fuel has been the biggest limiting factor in developing a global prediction system for fire activity. The datasets used here rely on a physical understanding of fuel dynamics to derive a consistent picture of fuel characteristics over time. The use of a physical model for fuel could be avoided if sufficient information was available directly from the observations[63]. For most applications direct observations are scarce, if available at all. For now, the creation of these fuel datasets is based on process-based models that are used to inform the final ML model[38]. A physical-derived dataset might remain a prerequisite for successful machine learning applications for many applications where the observing system is limited. Our findings indicate that the acquisition of high-quality global data is paramount to successfully train a ML-based fire activity model.

## Methods

### ML infrastructures

We use the data driven approach developed in the ECMWF model called the Probability of Fire (PoF)[36]. This model has been running in ECMWF since 2023 and produces daily forecasts available to ECMWF users. PoF predicts active fire (AF) observations from the MODIS MCD14ML active fire product (collection 6.1; 1 km resolution[26]). Fires flagged as low confidence in the AF product were not used. The PoF system uses gradient-boosted decision trees from the XGBoost library on detected active fires[64]. The training iteratively adds models to correct errors made by previous iterations, resulting in a computationally efficient optimization. The system training uses a classifier approach which defines a positive hit as an active fire detection within the grid cell on a given day.

To compare multiple data-driven approaches we also trained random forest and neural network models. The random forest model implemented using[65] consists of 50 trees with a maximum depth of 5, and each tree is trained on 80% of the samples using 50% of the features for each split. This configuration ensures a balance between model complexity and computational efficiency, while leveraging ensemble learning for robust performance.

The neural network model consisted of two hidden layers with 32 and 16 nodes, respectively, with sequential information flow from the input to the output layer using ReLU activation functions. The features were standardized. The model was trained using the Adam optimizer[66] with a learning rate of 0.001, categorical cross-entropy loss, and a batch size of 64 for up to 15 epochs. To prevent overfitting and optimize training efficiency early stopping was applied.

Several alternative configurations of each model was explored, including random forest models with 100 and 200 estimators and an neural network model with a four-layer architecture. These modifications yielded only marginal improvements, with the RF normalized average score increasing from 0.77 to 0.78 for both 100 and 200 estimators, and the NN from 0.83 to 0.84. Given the negligible performance gains relative to the increased computational cost, we opted to retain the simpler configurations for efficiency.

While tree-based methods like XGBoost and random forests lack temporal memory, this limitation is mitigated by including input features such as live fuel moisture content and fuel load which inherently include memory of previous conditions. For example, the deep soil moisture with a memory of several months. In contrast, neural networks have the potential to capture complex nonlinear interactions in the data but are computationally expensive, require extensive hyperparameter tuning and are less interpretable for probabilistic classification tasks. Performance-wise they are often shown to be outperform by XGBoost for tasks such as classification[67,68].

### Training dataset

We included 19 predictors of active fires that are grouped into three controls: weather, fuel and ignition (Table 1).

Weather variables are from ERA5-Land (9 km resolution[69]) Fuel variables are from the fuel characteristic model of McNorton and Di Giuseppe[38] also available at 9 km resolution. This model uses ESA-CCI above ground biomass estimates[70] and Copernicus Atmosphere Monitoring Service net ecosystem exchange estimates[71] to infer fuel abundance. Abundance is then split between live leaf and wood load,

and dead foliage and wood load based on the leaf area index (LAI) and vegetation type. Fuel moisture content is split between live fuel, dead foliage and dead wood. Live fuel moisture is a function of LAI, soil moisture and vegetation type whilst dead fuel is based on an extension of the Nelson model[72].

Ignition drivers include variables known to indirectly control human capability to ignite a fire such as population density, proximity to urban areas and road density. Lightning density is also included as a source of natural ignitions from ECMWF analysis[15,45,73].

Grouping the set of 19 drivers between the three controls which forms the three sides of the fire triangle—weather, fuel and ignitions—is not straightforward, as fuel moisture and weather variables are strongly correlated and fuel load is also related to weather conditions. Hence, some variables can be associated to more than one control. The clustering stems from the historical reason of preserving the same weather drivers as the classical fire danger indices. It also takes into consideration the different models that have been used to generate the datasets.

## Target dataset

We describe fire activity in terms of active fires. While a similar approach could be employed for estimating burned areas, the fire weather index (FWI) was specifically derived to characterize fire intensity, making it closer in nature to active fires, hence our choice. The longest coherent global observations come from the MODIS AQUA and TERRA satellites, launched in 1999 and 2002, respectively. The active fire product MCD14 v6.1 is the latest version of the dataset[21,74]. Although the observation time-series spans over two decades, they are still short compared to the decadal to centurial fire return intervals in many ecosystems. As consistent long-term records of fire extent and properties are fundamental for the presented analysis, the training would benefit for an extension of the record and also an increase in resolution. While fire observations from other moderate-resolution datasets (such as the visible infrared imaging radiometer suite (VIIRS)) and high-resolution datasets (e.g. Landsat and Sentinel sensors) are increasingly available they do not provide the long time record of MODIS. It is important to note that the accuracy of our predictions could be significantly enhanced with access to a more robust dataset from multiple sensors. Therefore, while the results provided here are very encouraging it is important that we take into account the potential for utilizing a broader dataset to improve the accuracy of fire activity predictions for an operational use.

## Validation metric

We use a set of metrics to assess the quality of the predictions. In the following we define what these are and why they were chosen.

**Correlation.** Correlation measures the linear relationship between the predicted and observed fire activity. Given we provide a forecast in terms of probability of occurrence and not the binary occurrence which is observed, the interpretation of the correlation is to be intended as the concomitant occurrence of high probabilities when fire events are observed. Given the sample is highly bias in favor of non-fire predictions, the correlation gives more weight to positive detection and predictions. A point to notice is that while correlation provides an overall estimations of how well high probability correspond to observed fires, in this specific application is a metric which has limitation as fires are events that occur at relatively low probabilities (1–2%) so values between prediction and fire activity tends to be very low. Most of the information here comes therefore from comparing across experiments rather than from looking at the absolute value.

**Brier score.** The Brier score is a measure of the accuracy of probabilistic predictions. It is calculated as the mean squared difference between the predicted probability and the actual outcome (0 or 1). The

Brier score is therefore a true probabilistic skill score as it evaluates the quality of the probability estimates. It considers both the calibration and the refinement of the predictions, making it a good fit for probabilistic forecasts where the probabilities are generally low.

**Logarithmic loss (Logloss).** Logloss measures the performance of a classification model by penalizing false predictions. It is calculated as the negative log of the likelihood of the true labels given the predicted probabilities. Logloss is sensitive to the confidence of the predictions. It strongly penalizes predictions that are confident and wrong, which is useful for ensuring that the model not only predicts the correct class but also assigns appropriate probabilities. In scenarios, such as fire prediction, where the observed outcomes are mostly zeros, Logloss remains informative because it considers the probability distribution of predictions. It is less biased by the imbalance, unlike some other metrics that might favor the majority class.

**Receiving operator curve (ROC) and area under the curve (AUC).** The ROC curve is a plot of the true positive rate (recall) against the false positive rate at various probability thresholds. The AUC shown in Fig. 2 represents the model's ability to distinguish between classes. The ROC–AUC is useful for evaluating unbalanced datasets, such as the datasets used here, as it considers all possible classification thresholds, providing a single value to summarize the model's performance across all thresholds. A perfect model would have an ROC–AUC of 1, while a model with any misclassifications will have an ROC–AUC below 1.

**Expected calibration error.** Expected calibration error (ECE) quantifies the reliability of predictions, specifically, how well the probabilities align with observed frequencies of fire occurrence. We assess the reliability by comparing predicted probabilities with observed fires over a range of probability bins, with a lower ECE indicating better calibration. For our dataset, where fire events are uncommon, the ECE is useful as it provides a measure of skill, even in the case of low-event probabilities. By optimizing for low ECE, we ensure that the model not only identifies potential fire occurrences but does so in a manner that reflects true likelihoods, without under- or over- predicting.

**Reliability.** The panels in Fig. 3 are based on a reliability skill score, which we have identified as the most relevant metric in our analysis. Reliability in fire forecasting refers to the accuracy with which a predictive model's forecasted probabilities match the actual observed frequency of fire events. Thus, it measures how well the predicted likelihood of fires corresponds to their real-world occurrence.

A reliable fire forecast model ensures that the probabilities it assigns to fire events are calibrated correctly. For example, if a model predicts a 0.3% probability of fire on a given day, then, on average, fire should occur on 0.3% of such days over a long period. We use annual estimates, as resource planning is typically performed on a yearly basis. Hence, a consistent 0.3% probability prediction is successful if it corresponds to 1 observed fire per year.

This concept is crucial because it determines the model's trustworthiness in practical applications, such as resource allocation for fire prevention and response. Moreover, it can be used to assess the increased probability of fire occurrence due to external factors like human practices or climate change.

Reliable fire forecasting is also important for minimizing false alarms, which can lead to unnecessary resource expenditure, and for avoiding missed warnings, which can result in unpreparedness and greater harm from unexpected fires.

A potential challenge in computing a reliability score aggregated over a large spatiotemporal domain is the issue of compensating biases. For instance, consistent over-predictions in one region may be offset by consistent under-predictions in another, leading to an

artificially high reliability score. A similar effect can occur when considering temporal biases. Therefore, analyzing regional reliability scores is crucial for accurately assessing model skill. Our analysis indicates that, for the models used in this study, compensating biases do not typically arise from seasonal variations in bias.

## Fire weather index

The Canadian fire weather index, derives from the four main inputs the vegetation moisture state representative of different depths of the forest floor[75]. It further expands on this by using wind and long-term precipitation deficit to derive a potential rate of spread index and a build-up indicator which then combined define a generic index of fire danger known as the fire weather index (FWI)[48]. A higher FWI indicates fire weather conditions more conducive to wildfires once ignited. The index is derived assuming a specific forest type, "Pinus Banksiana," and thus an environment with a sufficient fuel load. The FWI is especially useful for predicting the likelihood and severity of extreme events in ecosystems where weather is the primary limitation to fire (i.e., those mainly limited by moisture or temperature) because of its original design for use in forest ecosystems[76]. In areas with limited fuel[40] and where burning is heavily controlled by human practices[28], the correlation with actual fire activity is limited. FWI is extensively used in operational global information platforms such as the European Forest Fire Information System (EFFIS), the Global Wildfire Information System (GWIS), and the Canadian Wildland Fire Information System (CWFIS)[51,52,77,78].

To produce daily FWI data outlined in this study for 2023 at 9 km resolution, we used the global ECMWF Fire Forecast (GEFF) model 4.1 forced by ERA5-land data[69]. The FWI dataset is generated at 9 km resolution using the same forcing as the fuel model. FWI assesses potential fire danger by integrating key meteorological factors, specifically temperature, humidity, wind speed, and precipitation, providing a quantitative measure of landscape flammability. Most global[18,51] and regional[52] early warning systems employ this metric as a generic measure of fire danger.

We acknowledge that the FWI is not the only index for fire danger, and sub-indices of this system or other fire danger systems developed for other regions[50,79–81] can be used to improve the information provided to forestry agencies in biomes different from the boreal forest of Canada, for which the FWI was derived. The global performance of fire weather indices could, in absolute terms, be better than what is shown here. However, the scope of this study is to assess the enhanced ability in directly forecasting fires when a data-driven approach is applied to match observed fire activity, and how much of this ability stems from datasets that correctly describe the whole fire process.

## Data availability

The input meteorological data for training the data-driven model presented here is taken from the ERA5-Land dataset openly available through the Copernicus Climate Data store (https://doi.org/10.24381/cds.e2161bac). The fuel characteristic dataset is available through weblinks at https://doi.org/10.24381/378d1497. The MODIS active fire product was downloaded from the University of Maryland SFTP (formerly FTP) server. Connect using the following information: Server: fuoco.geog.umd.edu Login name: fire Password: burnt. Spurious signals were removed using a Copernicus Atmosphere Monitoring Service mask which can be requested to the corresponding authors. The data to create Figs. 1–4 are available as pickle files on zenodo https://zenodo.org/records/11653699. The full set of data-driven fire activity forecast for 2023 in the multiple configurations processed for this study are available through the corresponding authors. Daily operational data-driven model output for the best training configuration (images and WMS layers) is available to registered users through the ECMWF Web platform, ECCharts (https://eccharts.ecmwf.int).

## Code availability

The main scripts for data processing, model training, and analysis are archived in a publicly accessible repository https://doi.org/10.24433/CO.8570224.v1, with documentation to facilitate replication of the results.

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

## Acknowledgements
F.D.G. and J.M. are funded by the Copernicus Emergency Management Service contract no. 942604 between the Joint Research Centre and ECMWF. The discussion section has benefited from several informal conversations about the role of process-based and data-driven forecasts with colleagues at both ECMWF and ESA over the past years. We also acknowledge the discussion on the role of physical base fire science and the technological advancement of ML at the the GOFC-GOLD Fire Implementation meeting hosted in Canada in 2023. Credits: Graphical elements of Fig. 1 were designed by Freepik (https://www.freepik.com).

## Author contributions
F.D.G. and F.W. conceived the idea. J.M. implemented the experiments. F.D.G. and J.M. performed the data analysis. A.L. curated the graphical display and generated the infographic displayed in Fig. 1. F.D.G. wrote the paper. All authors contributed to the interpretation of the results and revised the manuscript.

## Competing interests
The authors declare no competing interests.
