## [Transparent Peer Review file · Nature Communications]

Global data-driven prediction of fire activity

Corresponding Author: Dr Francesca Di Giuseppe

Editorial Note: This manuscript has been previously reviewed at another journal. This document only contains information relating to versions considered at *Nature Communications*.

Version 0:

Reviewer comments:

Reviewer #1

(Remarks to the Author)

This research addresses an important topic in the field of fire forecasting, specifically the integration of machine learning (ML) methods with traditional fire weather indices to improve predictions of fire activity. The study highlights the significant role that global datasets and data quality play in enhancing forecasting skill. This focus is timely and relevant, given the increasing reliance on data-driven approaches in environmental sciences. However, despite the importance of this topic, I have significant concerns regarding the methodological justification and the completeness of the analysis presented in this paper.

1. I fully agree with Reviewer 2's original concern regarding the comparison between the process-based Fire Weather Index (FWI) and the data-driven model outputs. The authors have treated these as directly comparable, but as Reviewer 2 correctly pointed out, the data-driven outputs are not FWI but rather fire intensity metrics derived from MODIS active fire products. This indirect comparison is problematic because the two outputs fundamentally represent different metrics and assumptions. The authors' response to this issue does not convincingly justify the validity of this comparison. Moreover, the dependence of subsequent sensitivity tests on this flawed comparison undermines the credibility of the results. If the foundational comparison is not robust, the derived conclusions about the impact of fuel, ignition sources, and other factors are difficult to justify. Without a more rigorous justification or alternative methodology for comparison, this issue remains a significant weakness of the study. So, the authors should provide a more robust justification for the comparability of process-based and data-driven outputs. If such a justification is not feasible, they should consider revising their analysis to avoid direct comparisons or frame their results differently.

2. While the authors emphasize the pivotal role of data quality and availability in driving improvements in fire forecasting, they do not provide sufficient detail about the range of ML methods tested in their analysis. The current manuscript lacks a comparative analysis of different ML approaches and their sensitivities, which is a critical aspect of understanding the robustness and generalizability of their findings. Including a systematic evaluation of multiple ML methods (e.g., random forests, neural networks, gradient boosting) would provide valuable insights into the sensitivity of forecasting results to the choice of algorithm. Such an analysis would also help clarify whether the improvements observed are primarily due to data quality or whether certain ML methods inherently perform better for this application. Additionally, the authors could strengthen their argument by explicitly linking the sensitivity results of different ML methods to the combinations of weather, fuel, and ignition data they analyzed. This would offer a more comprehensive assessment of the methodological landscape and address a gap in the current study. So, a broader evaluation of ML methods should be included, with clear documentation of their sensitivity results. This would enhance the manuscript's methodological rigor and provide a stronger basis for the authors' conclusions regarding the importance of data quality versus algorithm choice.

(Remarks on code availability)

Reviewer #2

(Remarks to the Author)

NCOMMS-24-68236-T – Global data-driven prediction of fire activity – a Review

I have finished reviewing the paper "Global data-driven prediction of fire activity". The authors develop Machine Learning (ML) models to evaluate the probability of fire based on meteorological factors, fuel characteristics, and more. They compare

their predictions to those of the traditional Fire Weather Index (FWI) and highlight their higher performance. The technical quality of the manuscript appears sound. However, there are a few major issues and limitations that the authors must address.

First of all, the literature review is lacking. It appears that the authors are unaware of the vast literature in this field. They provide only three citations of previous studies harnessing the power of ML for wildfire predictions. There are actually hundreds of papers in this field, including both regional and global predictions of wildfire using ML (see references). While it is obviously unnecessary to address each and every one of these, the paper should acknowledge the main body of work in this field and highlight the relevant conclusions drawn from such efforts.

The second issue, which is related to the first one, is that the paper does not provide much scientific novelty. As can be seen from the references provided, the advantage of ML in wildfire predictions has been established exhaustively, even in global scales. The data described by the authors (e.g., fuel characteristics) is widely acknowledged by the wildfire science community. The Discussion section should highlight the novel scientific conclusions of the research.

An additional issue relates to one of the main conclusions in the paper: In their Discussion section, the authors imply that there is no need for more “sophisticated” algorithms, and that additional data is more important. While data is undoubtedly a central element in data-driven models, the suggestion that there is no need to improve the ML models is not established in the MS. The authors only developed a single ML model; it is possible that alternative models could have a higher performance, even with exactly the same data. The authors should either provide a comparison of various ML models, or remove this conclusion.

Additional comments:

“As a result, the two models predict slightly different quantities. However, the similarities between them are sufficient for a direct comparison.” – this requires further explanation.

The precision in Figs. 7-8 appears incredibly small (~0.01). How do the authors explain this?

The authors should also show common metrics such as accuracy and recall.

There are almost no details on the model used in the research, except for it being based on the XGB library. How were the hyperparameters chosen? Also, no code is provided in the submission to evaluate the model.

The authors mention Feature Importance based on the XGB or SHAP library in the Methods, but I could not find them in the manuscript.

Why don't the authors include humidity as a feature? This is known to be a central predictor of wildfire risk.

References:

Ji, Y., Wang, D., Li, Q., Liu, T., & Bai, Y. (2024). Global Wildfire Danger Predictions Based on Deep Learning Taking into Account Static and Dynamic Variables. *Forests*, 15(1), 216.

Prapas, I., Ahuja, A., Kondylatos, S., Karasante, I., Panagiotou, E., Alonso, L., ... & Papoutsis, I. (2022). Deep learning for global wildfire forecasting. *arXiv preprint arXiv:2211.00534*.

Shmuel, A., & Heifetz, E. (2023). Developing novel machine-learning-based fire weather indices. *Machine Learning: Science and Technology*, 4(1), 015029.

(Remarks on code availability)

Version 1:

Reviewer comments:

Reviewer #1

(Remarks to the Author)

Thank you for your thorough revisions and detailed responses to the reviewers' comments. I have reviewed the updated manuscript and the responses provided to the reviewers. It is clear that you have made significant efforts to address all previously raised concerns, and the added new analysis/prediction, including the Los Angeles fire case study and robustness tests using different machine learning methods, further enhances the manuscript. These additions provide valuable insights and strengthen the robustness of your findings. I find the revised manuscript to be well-structured and scientifically sound. I am satisfied with the improvements and have no further major concerns. Congratulations on an excellent revision!

(Remarks on code availability)

N/A

Reviewer #2

(Remarks to the Author)

NCOMMS-24-68236A – Global data-driven prediction of fire activity – a Review

I have finished reviewing the revised version of the paper “Global data-driven prediction of fire activity” (NCOMMS-24-68236A). Although the authors have made some improvements to the manuscript, I am still concerned about several issues. Most importantly, the authors did not properly address the question about novelty compared to previous studies. Their letter

only notes the application in a real-time system; while the application of an ML-based system is commendable, from my perspective it does not justify scientific novelty by itself. If the manuscript were to include some insights that could not be gained otherwise, this would justify the scientific novelty; however, unfortunately I currently see no such insights. More problematically, the manuscript itself still has a statement which I do not believe to be accurate:

“Machine learning is already widely applied in multiple areas of fire science, including fire management practices [30], early [31] and long term detection [32], and crucially to aid firefighting operations on a very local scale [33–35]. Still, expanding fire prediction to a global scale and testing the limits of AI remains one of the most ambitious challenges in the field.” – the authors claim that so far ML applications have been limited to a very local scale, and it remains a gap in the literature to apply them on a global scale. This, despite the fact that my previous review included 3 examples of global scale wildfire predictions using ML.

An additional paper applying ML models on a global scale has also recently been published in Science:

Jones, M. W., Veraverbeke, S., Andela, N., Doerr, S. H., Kolden, C., Mataveli, G., ... & Abatzoglou, J. T. (2024). Global rise in forest fire emissions linked to climate change in the extratropics. *Science*, 386(6719), ead15889.

Regarding the conclusion “We evaluate how skillful this completely data-centric system has been, and determine where this predictability comes from—whether it is the training data or the data-driven approach itself.” – I still can’t understand this comment. Obviously both the data and the model are crucial for a data-driven approach; what does this conclusion actually mean? The authors themselves, in the revised version, show a difference between different ML models.

The advantage of XGBoost over Neural Networks is not due to the NN’s high complexity. The advantage of GBM models over NNs in tabular data is well acknowledged, has been studied extensively on large benchmarks, and has several potential explanations. See for example:

Grinsztajn, L., Oyallon, E., & Varoquaux, G. (2022). Why do tree-based models still outperform deep learning on typical tabular data?. *Advances in neural information processing systems*, 35, 507-520.

Regarding the very low precision (~0.01), I am not satisfied with the authors’ response. The authors themselves write in the manuscript: “For example, a model false alarm could result in investing resources when not needed”. A precision of 0.01 would mean that for every true prediction of the model, 100 alerts would be provided. This is a very high ratio, even in unbalanced data such as that of wildfires.

In my previous review I suggested the authors explain how hyperparameter optimization is performed. The authors have not responded fully, and now there is a similar issue with the new models added. Why have the authors chosen a 2-layer neural network? Also, the limited use of 50 estimators in the Random Forest model is extremely small compared to most studies. Have the authors evaluated the number of estimators and found no improvement above 50? There are many tools to assist hyperparameter optimization, including AutoML libraries. Here, it seems that the authors have chosen a neural network / RF model at random.

Have the authors removed the performance tables from the previous version?

(Remarks on code availability)

No code provided for review.

Version 2:

Reviewer comments:

Reviewer #2

(Remarks to the Author)

I have finished reviewing the revised version of the paper “Global data-driven prediction of fire activity” (NCOMMS-24-68236B). The authors have done a very good job in answering my previous comments, and I believe the manuscript has substantially improved. I have no further comments or suggestions.

I thank the authors for their hard work and their contribution to this important field.

(Remarks on code availability)

Reviewer #1 (Remarks to the Author):

This research addresses an important topic in the field of fire forecasting, specifically the integration of machine learning (ML) methods with traditional fire weather indices to improve predictions of fire activity. The study highlights the significant role that global datasets and data quality play in enhancing forecasting skill. This focus is timely and relevant, given the increasing reliance on data-driven approaches in environmental sciences. However, despite the importance of this topic, I have significant concerns regarding the methodological justification and the completeness of the analysis presented in this paper.

We thank the reviewer for finding the topic of this work highly relevant. In today's landscape, where there is growing trust in the possibilities of machine learning (ML), it is crucial to understand the origins of predictions. We have addressed the reviewer's two main concerns and have substantially reframed the study to emphasize the critical role of data versus the complexity of the ML framework.

To this end, we conducted ablation experiments using two additional infrastructures, as suggested by the reviewer: random forests (RF) and neural networks (NN). By adding a low-complexity ML architecture (RF) and a high-complexity one (NN) to the mid-complexity architecture (XGBoost), we demonstrate that, once the algorithm is fit for purpose, increasing the network's complexity yields little returns. Instead, greater improvements can be achieved by incorporating data that fully characterizes the process being modelled.

The analysis comparing our model to the process-based fire weather index (FWI) has been relegated to a secondary focus. One of the primary criticisms of this analysis was the inability to perform a quantitative skill comparison between AI-based fire forecasts, trained on active fire data, and FWI. The reviewer suggested either removing or rephrasing this analysis. In response, we have removed the quantitative analysis and the associated Figure 1. Instead, we present a qualitative analysis comparing the expected climatological probabilities of fire occurrence with FWI values to better align with fire activity climatology.

In addition, considering the shocking events recently witnessed in Los Angeles, we have included an analysis of these events in the paper since it highlighted the evident advantage of using PoF. While FWI forecasts indicated very extreme fire danger across all municipal areas in Los Angeles, PoF forecasts provided much better localization of the Palisades and Eaton fires. We highlight how this improved localization stems from the inclusion of fuel load data in the formulation, reflecting the exceptionally high fuel loads due to the wet spring. The new sections reads:

Starting on January 7, 2025, a series of catastrophic wildfires devastated the Los Angeles metropolitan area and surrounding regions. These fires were fueled by extremely low humidity, dry conditions, and hurricane-force Santa Ana winds, which reached speeds of 160 km/h in some areas. The wildfires claimed several lives, destroyed or damaged thousands of residential and commercial structures, and forced nearly 200,000 people to evacuate. The most severe impacts were caused by the two largest fires: the Palisades Fire and Eaton Fires, which remained uncontrolled for days. Traditional fire weather indices had indicated that persistent anomalous and dry conditions for the winter season and the incoming katabatic winds would create ideal anomalous conditions and warnings had been issued over the regions. However, FWI indicated widespread danger conditions (Figure 5). The data-driven model including all the variables shows better localization of areas where fires ignited (Figure 4). This better skill stems from the inclusion of information on fuel abundance and fuel status. In fact, the severity of the fires was linked to a phenomenon known as "hydroclimate

whiplash" (Swain et al 2025) which is believed to be exacerbated by climate change. This phenomenon occurs when anomalous wet conditions are followed by very dry conditions. The wet spell promotes vegetation growth and a rapid increase in fuel availability. During the subsequent dry period, moisture is quickly depleted, leaving the fuel highly flammable and often resulting in severe burning. California experienced two very wet spring seasons preceding the 2025 fires, which contributed to an enhanced availability of fuel in the region. These anomalies created a distinct pattern of increased flammability at the wildland-urban interface. This pattern was effectively captured by the ML model, which accounted for the anomalous fuel conditions but remained undetected by traditional FWI metrics. We believe that the revised analysis highlight the transition that AI facilitates—from forecasting fire danger to actually forecasting fire activity and how this could really benefit people on the ground by reducing the many false alarms of traditional methods.

1. I fully agree with Reviewer 2's original concern regarding the comparison between the process-based Fire Weather Index (FWI) and the data-driven model outputs. The authors have treated these as directly comparable, but as Reviewer 2 correctly pointed out, the data-driven outputs are not FWI but rather fire intensity metrics derived from MODIS active fire products. This indirect comparison is problematic because the two outputs fundamentally represent different metrics and assumptions. The authors' response to this issue does not convincingly justify the validity of this comparison. Moreover, the dependence of subsequent sensitivity tests on this flawed comparison undermines the credibility of the results. If the foundational comparison is not robust, the derived conclusions about the impact of fuel, ignition sources, and other factors are difficult to justify. Without a more rigorous justification or alternative methodology for comparison, this issue remains a significant weakness of the study. So, the authors should provide a more robust justification for the comparability of process-based and data-driven outputs. If such a justification is not feasible, they should consider revising their analysis to avoid direct comparisons or frame their results differently.

We have substantially reworked this section, removing the quantitative comparison. While we still believe that comparing our approach with traditional methods is important, we understand that the direct comparison in terms of correlation distracted readers from the main message of the paper, which focuses on identifying the real drivers of a potential ML revolution in fire forecasting.

Following the reviewer's suggestion, we have rethought the comparison and included a qualitative analysis instead. A completely new section titled *Predicting Fire Activity Rather than Fire Danger* has been added. The original Figure 1 has been removed, and a new figure, previously part of the supplementary material, has been incorporated into the main text. There are substantial revisions (in track changes) that implements the reviewer suggestion. We thank the reviewer for suggesting this way forward and believe the paper has really gained from this suggestion.

2. While the authors emphasize the pivotal role of data quality and availability in driving improvements in fire forecasting, they do not provide sufficient detail about the range of ML methods tested in their analysis. The current manuscript lacks a comparative analysis of different ML approaches and their sensitivities, which is a critical aspect of understanding the robustness and generalizability of their findings. Including a systematic evaluation of multiple ML methods (e.g., random forests, neural networks, gradient boosting) would provide valuable insights into the sensitivity of forecasting results to the choice of algorithm. Such an analysis would also help clarify whether the improvements observed are primarily due to data quality or whether certain ML methods inherently perform better for this application.

Additionally, the authors could strengthen their argument by explicitly linking the sensitivity results of different ML methods to the combinations of weather, fuel, and ignition data they analyzed. This would offer a more comprehensive assessment of the methodological landscape and address a gap in the current study. So, a broader evaluation of ML methods should be included, with clear documentation of their sensitivity results. This would enhance the manuscript's methodological rigor and provide a stronger basis for the authors' conclusions regarding the importance of data quality versus algorithm choice.

We have followed the reviewer's suggestion and added an analysis comparing less and more complex infrastructures. The findings suggest that while there is a clear benefit in increasing complexity from a simple random forest to XGBoost, the performance gain levels off when moving to more complex architectures, such as neural networks (NN).

In contrast, the role of data is extremely valuable. Even with a simpler architecture, incorporating fuel-related information into the training significantly enhances the model's predictive capability. Figure 1 now includes an overview of the results from the additional ML methods, and we have incorporated these two methods into the methods section.

Reviewer #2 (Remarks to the Author):

NCOMMS-24-68236-T – Global data-driven prediction of fire activity – a Review

I have finished reviewing the paper “Global data-driven prediction of fire activity”. The authors develop Machine Learning (ML) models to evaluate the probability of fire based on meteorological factors, fuel characteristics, and more. They compare their predictions to those of the traditional Fire Weather Index (FWI) and highlight their higher performance. The technical quality of the manuscript appears sound. However, there are a few major issues and limitations that the authors must address.

We would like to thank the reviewer for the overall positive comments to our work. We have addressed the suggested revision and substantially restructure the manuscript while maintaining the main message. Detailed answers to the concerns raised is provided in the following

First of all, the literature review is lacking. It appears that the authors are unaware of the vast literature in this field. They provide only three citations of previous studies harnessing the power of ML for wildfire predictions. There are actually hundreds of papers in this field, including both regional and global predictions of wildfire using ML (see references). While it is obviously unnecessary to address each and every one of these, the paper should acknowledge the main body of work in this field and highlight the relevant conclusions drawn from such efforts.

We apologize for omitting some of the references. We had initially chosen to reference reviews on the subject, such as Jain (2020), rather than providing a lengthy list of individual works. Additionally, we may have misinterpreted the Nature Portfolio guidelines regarding the inclusion of open archive publications. We have now revised and extended this section to include the suggested citations and some very recent papers that have been just coming out.

The second issue, which is related to the first one, is that the paper does not provide much scientific novelty. As can be seen from the references provided, the advantage of ML in wildfire predictions has been established exhaustively, even in global scales. The data described by the authors (e.g., fuel characteristics) is widely acknowledged by the wildfire science community.

There has indeed been several analyses exploring the potential to predict certain aspects of fire using data-driven approaches, however, this study is the first to implement an operational fire forecasting system based on data-driven technology. It also focuses on an innovative aspect: the relative importance of various information sources and, ultimately, identifying the true drivers of the potential ML revolution in fire forecasting. We argue that the key lies in the **quality and diversity of the data** sources rather than the complexity of the ML algorithms themselves. We have made this clearer in the paper.

Moreover, this work is built on an operational system running in real time, delivering daily forecasts that are actively used by numerous institutional users. The usability of this system enabled us to perform an on-the-fly analysis of the very recent extreme event in California, which occurred just a few days ago and is still ongoing. This event provided us with a unique opportunity to demonstrate, through real-world application, the origins and accuracy of the predictions. The new section in the results titled California 2025 reports on these new results.

The Discussion section should highlight the novel scientific conclusions of the research. An additional issue relates to one of the main conclusions in the paper: In their Discussion section, the authors imply that there is no need for more “sophisticated” algorithms, and that additional data is more important. While data is undoubtedly a central element in data-driven models, the suggestion that there is no need to improve the ML models is not established in the MS. The authors only developed a single ML model; it is possible that alternative models could have a higher performance, even with exactly the same data. The authors should either provide a comparison of various ML models, or remove this conclusion.

We have followed the reviewer’s suggestion and added an analysis comparing less and more complex infrastructures. The findings suggest that while there is a clear benefit in increasing complexity from a simple random forest to XGBoost, the performance gain levels off when moving to more complex architectures, such as neural networks (NN).

In contrast, the role of data is extremely valuable. Even with a simpler architecture, incorporating fuel-related information into the training significantly enhances the model’s predictive capability. Figure 1 now includes an overview of the results from the additional ML methods, and we have incorporated these two methods into the methods section. The addition of this analysis confirms the statement in the conclusions.

Additional comments:

“As a result, the two models predict slightly different quantities. However, the similarities between them are sufficient for a direct comparison.” – this requires further explanation.

Based on the editor’s and reviewers’ suggestions, we’ve reworked this section quite a bit, removing the quantitative comparison. While we still think comparing our approach to traditional methods is valuable, we understand that the correlation-based comparison was not very clear and shifted the focus away from the main message of the paper—identifying the real drivers behind a potential ML revolution in fire forecasting.

Taking the reviewers’ advice, we have rethought the comparison and switched to a qualitative analysis instead. We have added a new section called *Predicting Fire Activity Rather than Fire Danger* and replaced the original Figure 1 with a new figure that was previously in the supplementary material.

We really appreciate the reviewers’ suggestions, as we feel these changes have made the paper much stronger.

The precision in Figs. 7-8 appears incredibly small (~0.01). How do the authors explain this?

The low precision score reflects class imbalance of the problem or the rarity of fires, not necessarily poor model performance.

The authors should also show common metrics such as accuracy and recall.

While recall measures the ability of the model to identify true positives, probabilistic metrics like the Brier score already evaluate the closeness of predicted probabilities to true outcomes, effectively accounting for missed events using the raw probabilities. Recall, however, relies on a threshold to categorise the predictions as positive or negative. Given that our model produces low probabilities (typically well below 5%), the selection of such a threshold is subjective and may not fully reflect the models reliability. Similarly, accuracy also

assumes binary classification and does not directly evaluate the quality of probabilistic forecasts, making it less suitable for our analysis.

There are almost no details on the model used in the research, except for it being based on the XGB library. How were the hyperparameters chosen? Also, no code is provided in the submission to evaluate the model.

Full details of the model and hyperparameters are detailed in the referenced article (McNorton et al., 2024).

The authors mention Feature Importance based on the XGB or SHAP library in the Methods, but I could not find them in the manuscript.

Although in this study we do not present feature importance or the use of the SHAP library the comment was intended to reference potential use of the XGBoost architecture. We agree this was not made clear and have now removed this statement as it is not helpful in this context

Why don't the authors include humidity as a feature? This is known to be a central predictor of wildfire risk.

Surface Humidity is included as a predictor in the form of dew point temperature (see figure 6)

References:

Ji, Y., Wang, D., Li, Q., Liu, T., & Bai, Y. (2024). Global Wildfire Danger Predictions Based on Deep Learning Taking into Account Static and Dynamic Variables. *Forests*, 15(1), 216.

Prapas, I., Ahuja, A., Kondylatos, S., Karasante, I., Panagiotou, E., Alonso, L., ... & Papoutsis, I. (2022). Deep learning for global wildfire forecasting. *arXiv preprint arXiv:2211.00534*.

Shmuel, A., & Heifetz, E. (2023). Developing novel machine-learning-based fire weather indices. *Machine Learning: Science and Technology*, 4(1), 015029.

These reference and also others have been included

REVIEWER COMMENTS

Reviewer #1 (Remarks to the Author):

Thank you for your thorough revisions and detailed responses to the reviewers' comments. I have reviewed the updated manuscript and the responses provided to the reviewers. It is clear that you have made significant efforts to address all previously raised concerns, and the added new analysis/prediction, including the Los Angeles fire case study and robustness tests using different machine learning methods, further enhances the manuscript. These additions provide valuable insights and strengthen the robustness of your findings. I find the revised manuscript to be well-structured and scientifically sound. I am satisfied with the improvements and have no further major concerns. Congratulations on an excellent revision!

Thank you once again for helping sharpening our message and the supportive and constructive review

The Code ocean capsule has been approved and available at

<https://codeocean.com/capsule/8570224/tree>

Reviewer #2 (Remarks to the Author):

NCOMMS-24-68236A – Global data-driven prediction of fire activity – a Review

I have finished reviewing the revised version of the paper “Global data-driven prediction of fire activity” (NCOMMS-24-68236A). Although the authors have made some improvements to the manuscript, I am still concerned about several issues.

Most importantly, the authors did not properly address the question about novelty compared to previous studies. Their letter only notes the application in a real-time system; while the application of an ML-based system is commendable, from my perspective it does not justify scientific novelty by itself. If the manuscript were to include some insights that could not be gained otherwise, this would justify the scientific novelty; however, unfortunately I currently see no such insights.

Thank you for your comments, we will try to answer them below

Regarding the novelty, we do think the paper brings new insights on fire modelling and forecasting, especially in an operational sense. The technology we describe provides global active fire predictions at a 9 km resolution. Several papers have explored the potential of machine learning (ML) for specific tasks, such as predicting FRP and, more commonly, burned areas. ML-based applications are also being used to support first response efforts at very local scales. However, none of these studies have actually replaced traditional fire weather forecasting systems in real-world applications, as the system we are analysing does.

There is also now a substantial body of literature employing ML to explain shifts in fire regimes, such as the recent *Science* paper by Matt Jones. However, these studies address a completely different problem: they aim to explain past and observed shifts in fire regimes rather than investigate where the predictive skill of a forecast originates. While the tools used

may be similar, the purpose is distinct. Indeed, Jones's work focuses on burned areas aggregated over vast biomes, rather than real-time high resolution fire forecasting.

The PoF infrastructure has also been applied within this past-scenario framework, other than for operational forecasting. An example of such an analysis is the *State of Wildfire 2023-2024* report, also led by Matt Jones last year, which extensively used PoF to trace back the most likely causes of observed fire activity.

However, this is not the focus of our paper. We aim to understand why the described system performs so well in real-time forecasting at an unprecedented resolution—predicting the future rather than explaining the past. We highlight the crucial role of high-quality data in driving this system, demonstrating that such accuracy is achievable globally at high resolution, albeit with significant computational resources. Our findings suggest that data quality is just as important as the ML framework itself and, most notably, that weather is not the highest-ranking factor among predictive variables.

To our knowledge, no study has addressed this question—indeed, no such forecasting system has been operating for a year, providing this type of real-time fire activity forecast.

The reviewer has not cited any work that addresses the problem of high-resolution fire activity prediction in an operational context or otherwise. The referenced literature consists of studies that have either attempted to reproduce burned areas at a much coarser resolution (25 km) or focused on fire driver attribution over large regions. Therefore, we believe we are not overlooking any major research on this topic.

Finally, we consider our study both highly important and particularly timely, given the system's strong performance in the recent Los Angeles case.

More problematically, the manuscript itself still has a statement which I do not believe to be accurate:

“Machine learning is already widely applied in multiple areas of fire science, including fire management practices [30], early [31] and long term detection [32], and crucially to aid firefighting operations on a very local scale [33–35]. Still, expanding fire prediction to a global scale and testing the limits of AI remains one of the most ambitious challenges in the field.” – the authors claim that so far ML applications have been limited to a very local scale, and it remains a gap in the literature to apply them on a global scale. This, despite the fact that my previous review included 3 examples of global scale wildfire predictions using ML.

We acknowledge our bias towards operational real-time forecasting, as this is what we consider a novel and important application. We have rephrased the text to explicitly highlight operational prediction of active fires in contrast to exploratory studies of burned areas.

Additionally, “local scale” here specifically refers to aiding firefighting operations rather than encompassing the entire body of literature. The sentence has been rephrased as following:

Global proof-of-concept AI predictive systems have mostly focused on reproducing burned areas. However, understanding the feasibility of a global, kilometre-scale fire activity prediction system is still in its early stages. An analysis of the skill of such a system, its limitations, and the key factors driving its performance is still lacking.

An additional paper applying ML models on a global scale has also recently been published in Science:

Jones, M. W., Veraverbeke, S., Andela, N., Doerr, S. H., Kolden, C., Mataveli, G., ... & Abatzoglou, J. T. (2024). Global rise in forest fire emissions linked to climate change in the extratropics. *Science*, 386(6719), ead15889.

We thank you for this and are aware of the study. This paper examines fire from a historical perspective rather than a forecasting perspective. Jones et al. applied a machine learning approach to analyse the drivers of fire in forest ecoregions, categorised them into 12 global forest pyromes, and described their varying sensitivities to climate, human influence, and vegetation. Their analysis is based on burned area and does not focus on forecasts. Given this, we do not see its relevance to our study beyond the application of machine learning.

Regarding the conclusion “We evaluate how skillful this completely data-centric system has been, and determine where this predictability comes from—whether it is the training data or the data-driven approach itself.” – I still can’t understand this comment. Obviously both the data and the model are crucial for a data-driven approach; what does this conclusion actually mean? The authors themselves, in the revised version, show a difference between different ML models.

The reviewer notes: “Obviously both the data and the model are crucial for a data-driven approach; what does this conclusion actually mean?” However, this was not so obvious to us—rather, it is the central research question that the paper seeks to address, as clearly stated in the introduction.

The core argument of the paper is to quantify what the reviewer summarises as “both the data and the model are crucial.” Our conclusion is substantiated by a set of ablation experiments and is more nuanced than what might initially seem obvious. Once an appropriate infrastructure is in place, our findings show that the quality and suitability of the data have a greater impact than the complexity of the machine learning model. This is a crucial point for us, as it underscores where we believe future efforts should be directed.

The advantage of XGBoost over Neural Networks is not due to the NN’s high complexity. The advantage of GBM models over NNs in tabular data is well acknowledged, has been studied extensively on large benchmarks, and has several potential explanations. See for example:

Grinsztajn, L., Oyallon, E., & Varoquaux, G. (2022). Why do tree-based models still outperform deep learning on typical tabular data?. *Advances in neural information processing systems*, 35, 507-520.

In the Methods section, we acknowledge that XGBoost is widely recognised as one of the best methods for classification tasks, which is why it was ultimately selected for the ECMWF forecasting system. This is explicitly stated in the text, and we have now incorporated the suggested citation.

Performance-wise they are often shown to be outperform by XGBoost for tasks such as classification

Regarding the very low precision (~0.01), I am not satisfied with the authors’ response. The authors themselves write in the manuscript: “For example, a model false alarm could result in investing resources when not needed”. A precision of 0.01 would mean that for every true prediction of the model, 100 alerts would be provided. This is a very high ratio, even in unbalanced data such as that of wildfires.

We have carefully assessed the use of precision as a metric and find it unsuitable for our application. The reviewer correctly highlights the low precision scores, which are expected given the nature of the problem. Unlike the studies referenced by the reviewer, our model predicts active fire probabilities rather than other fire related metrics, it operates globally rather than scene-specifically, and functions at a much finer spatiotemporal resolution.

A low precision score is an inherent consequence of predicting rare events. For example, in a 10x10 grid cell domain over several days, high fire risk may result in only a single fire detection. While we would still consider the model to have skill in such cases, precision does not reflect this. Additionally, the claim that the model produces excessive false alarms is misleading, as only one warning would be provided in such a scenario, not 100+.

A fundamental issue with precision is its dependence on a fixed probability threshold, often set at 50%. Since our model probabilities rarely exceed this level, precision becomes an unreliable measure. While an alternative threshold could be defined, this introduces subjectivity and interpretation bias, making precision an inappropriate metric for evaluating our system.

For these reasons we have decided to substitute the precision with Expected Calibration Error, which directly quantifies the reliability of the predicted probabilities. Given that our model outputs probabilistic fire risk rather than binary classifications, calibration is the most meaningful assessment of skill. ECE allows us to evaluate whether a predicted probability, such as 5%, corresponds to fire occurrence 5% of the time—ensuring that the model provides well-calibrated and trustworthy probability estimates.

The ECE is now also a score featured among the metrics that determine the overall scoring of the system. As a result, the relative numbers in the triangle have slightly changed to reflect this substitution. These changes do not affect the conclusions.

In the method section we have added the description of the ECE skill score as follows:

Expected Calibration Error

Expected Calibration Error (ECE) quantifies the reliability of predictions, specifically, how well the probabilities align with observed frequencies of fire occurrence. We assess the reliability by comparing predicted probabilities with observed fires over a range of probability bins, with a lower ECE indicating better calibration.

For our dataset, where fire events are uncommon, the ECE is useful as it provides a measure of skill, even in the case of low-event probabilities. By optimizing for low ECE, we ensure that the model not only identifies potential fire occurrences but does so in a manner that reflects true likelihoods, without under- or over- predicting.

In my previous review I suggested the authors explain how hyperparameter optimization is performed. The authors have not responded fully, and now there is a similar issue with the new models added. Why have the authors chosen a 2-layer neural network? Also, the limited use of 50 estimators in the Random Forest model is extremely small compared to most studies. Have the authors evaluated the number of estimators and found no improvement above 50? There are many tools to assist hyperparameter optimization, including AutoML libraries. Here, it seems that the authors have chosen a neural network / RF model at random.

We appreciate the reviewer's comments regarding hyperparameter optimization and model selection. Below, we provide further clarification on our approach.

Our original XGboost model was manually tuned, and we found that multiple hyperparameter configurations resulted in minimal changes in performance. Based on this experience, we followed a similar approach for the RF and NN models, manually evaluating key hyperparameters to balance predictive skill and computational efficiency.

For the RF model, we tested an increased number of estimators (50, 200, and 300). While there was a small improvement in the normalized average score for the model trained on all features (fuel, weather and ignition/suppression), from 0.77 with 50 estimators to 0.78 with 100 and 0.78 with 200, the computational cost increased significantly, making higher values impractical for our large-scale application. Given this, we kept 50 estimators as it made training multiple models with different input features computationally viable.

Similarly, for the NN model, we evaluated deeper architectures, increasing the number of layers from 2 to 4. While this resulted in a marginal improvement in the normalized average score, from 0.83 to 0.84, it also led to a substantial increase in training time. Since the added complexity did not translate into meaningful gains in predictive performance, we opted for the simpler two-layer architecture when training the multiple versions of the NN model.

While AutoML tools are available for hyperparameter optimization, our results indicate that further tuning beyond our selected configurations would provide diminishing returns, especially when considering the noticeable model improvement brought about by considering new features. Our primary goal is to ensure that our models remain well-calibrated and reliable rather than to optimize for marginal gains at the cost of computational efficiency and believe the overall message regarding model improvement being most dependent on input features remains regardless of the optimised hyperparameters.

Please note that the information about hyper tuning have now been included in the text as follows

“Several alternative configurations of each model was explored, including random forest models with 100 and 200 estimators and an neural network model with a four-layer architecture. These modifications yielded only marginal improvements, with the RF normalized average score increasing from 0.77 to 0.78 for both 100 and 200 estimators, and the NN from 0.83 to 0.84. Given the negligible performance gains relative to the increased computational cost, we opted to retain the simpler configurations for efficiency.”

Have the authors removed the performance tables from the previous version?

The table was in supplementary material. We have now followed the usual convention and crated a separate file for these information. The table have been emended to include the new skill score

Reviewer #2 (Remarks on code availability):

The Code ocean capsule has been approved and available at

<https://codeocean.com/capsule/8570224/tree>